# Bayesian Federated Estimation of Causal Effects from Observational Data

Thanh Vinh Vo[1]          Young Lee[2]          Trong Nghia Hoang[3]          Tze-Yun Leong[1]

[1]School of Computing, National University of Singapore
[2]Harvard University
[3]School of Electrical Engineering and Computer Science, Washington State University

## Abstract

We propose a Bayesian framework for estimating causal effects from federated observational data sources. Bayesian causal inference is an important approach to learning the distribution of the causal estimands and understanding the uncertainty of causal effects. Our framework estimates the posterior distributions of the causal effects to compute the higher-order statistics that capture the uncertainty. We integrate local causal effects from different data sources without centralizing them. We then estimate the treatment effects from observational data using a non-parametric reformulation of the classical potential outcomes framework. We model the potential outcomes as a random function distributed by Gaussian processes, with defining parameters that can be efficiently learned from multiple data sources. Our method avoids exchanging raw data among the sources, thus contributing towards privacy-preserving causal learning. The promise of our approach is demonstrated through a set of simulated and real-world examples.

## 1 INTRODUCTION

Causal effect estimation is important in many real-life situations. For example: What is the effect of war in a specific region on world food supply? How would the blood pressure of a patient change if that patient took a new drug? How does coronary heart disease affect age- and gender-specific mortality rates? These questions are common in many areas, including personalized medicine [Powers et al., 2018], digital experiments [Taddy et al., 2016], political science [Green and Kern, 2012], etc., and especially recent events in the Covid-19 pandemic and the war of Ukraine. In practice, the relevant data essential for accurate and meaningful causal inference may reside in multiple, decentralized data sources

which cannot be shared or combined due to geographical, organizational, process, and/or privacy constraints. Some alternative solutions such as establishing data use agreements or creating secure data environments may not be possible and are often not easily implemented. In addition, it is important to know whether the causal estimands are reliable. Thus, estimating a confidence interval of the relevant causal effect together with its point estimates would give helpful insights into the uncertainty of the causal estimand. For example, a narrow confidence interval for individual treatment effect of smoking on lung cancer, where zero falls outside the confidence interval, means that the patient is at a higher risk of getting cancer.

Most of the recent causal effect estimators, e.g., Louizos et al. [2017], Shalit et al. [2017], Madras et al. [2019], are point estimates without considering the uncertainty of the causal estimands. Bayesian approaches, e.g., Imbens and Rubin [1997], Daniels et al. [2012], Talbot et al. [2015], Gutman et al. [2018], Ning et al. [2019], on the other hand, aim to learn the posterior distributions of the causal estimands to obtain higher-order statistics that capture the uncertainty. To derive these posterior distributions of the causal estimands, however, most, if not all of the existing efforts involve pooling the distributed data from multiple sources centrally to compute the model *marginal* likelihood, thus violating the privacy constraints mentioned above.

We propose a Bayesian framework that can learn the causal effects of interest without combining data sources to a central site, and, at the same time, learn higher-order statistics of the causal effects to understand their uncertainty. This federated learning approach [McMahan et al., 2017] has not been well studied for causal inference. Our contributions are summarized as follows:

- We propose the Federated Causal Inference (FedCI) [1] framework that fuses federated learning and causal inference to incorporate multiple data sources while maintain-

[1]Source code: `https://github.com/vothanhvinh/FedCI`.

*Accepted for the 38th Conference on Uncertainty in Artificial Intelligence* (UAI 2022).

ing the sources at their local sites.

- FedCI generalizes the Bayesian imputation approach [Imbens and Rubin, 2015] to a more generic model based on Gaussian processes (GPs); the resulting model is decomposed into multiple components, each of which handles a distinct data source.
- FedCI minimizes information transmitted among the sources, thus enabling privacy-preserving causal inference. The framework could support multiparty computation and differential privacy in future.

- We propose a variational approximation scheme for the proposed model, whose evidence lower bound can be decomposed additively across different data sources. This allows the parameters to be optimized via federated gradient averaging. We then leverage the computed predictive distribution to efficiently estimate the desired treatment effect quantities.
- We empirically evaluate the proposed framework on benchmark datasets, and show its competitive performance as compared to the recent baseline approaches trained on the combined datasets.

## 2 RELATED WORK

**Causal inference.** In most causal inference literature, the estimation of causal effects is performed directly on accessible local data sources. Hill [2011], Alaa and van der Schaar [2017, 2018] proposed nonparametric approaches to estimate causal effects. A growing literature, including Shalit et al. [2017], Yoon et al. [2018], Yao et al. [2018], Künzel et al. [2019], Nie and Wager [2021], used parametric methods to model the potential outcomes. Louizos et al. [2017], Madras et al. [2019] used the formulation of Pearl [1995] to estimate causal effects under the existence of latent confounding variables. Bica et al. [2020a,b] formalized potential outcomes for temporal data with observed and unobserved confounding variables to estimate counterfactual outcomes for treatment plans. Imbens and Rubin [1997], Daniels et al. [2012], Talbot et al. [2015], Gutman et al. [2018], Ning et al. [2019] are typical Bayesian methods that learn posterior distributions of the causal estimands. All these works were not proposed for the context of multi-source data which cannot be shared and combined as a unified dataset. Our model, in contrast, learns treatment effects while preserving the source data at their local sites. It is different from the problem of transportability of causal relations [e.g., Pearl and Bareinboim, 2011, Bareinboim and Pearl, 2013b,a, 2016, Lee et al., 2020], where theoretical tools were developed to transport causal effects from a source population to a target population, which does not take into account the above data privacy constraint.

**Federated learning.** Federated learning aims to train an algorithm across multiple decentralized clients, thus respect the privacy information of the data [McMahan et al.,

2017]. Federated stochastic gradient descent [Shokri and Shmatikov, 2015] and federated averaging [McMahan et al., 2017] are two variations of federated learning. Recent developments of federated learning, e.g., Álvarez et al. [2019], Zhe et al. [2019], de Wolff et al. [2020], Joukov and Kulić [2020], Sattler et al. [2019], Mohri et al. [2019] are formalized for a typical classification or regression problem. Also, it has recently been applied in facilitating multi-institutional collaborations without sharing patient data [Rieke et al., 2020, Sheller et al., 2020] and healthcare informatics [Lee and Shin, 2020, Xu et al., 2021]. Other biomedical applications of federated learning include predicting adverse drug reactions [Choudhury et al., 2019], stroke prevention [Ju et al., 2020], mortality prediction [Vaid et al., 2020], predicting outcomes in SARS-COV-2 patients [Flores et al., 2021], etc. Ng and Zhang [2022], Gao et al. [2021] are noticeable works that estimate causal graphs in federated setting, which is different from our work in estimating casual effects.

Following some recent works [e.g., Shalit et al., 2017, Yao et al., 2018, Oprescu et al., 2019, Künzel et al., 2019, Nie and Wager, 2021], we develop a federated causal inference algorithm based on the potential outcomes framework. We summarize the related models in the subsequent sections.

### 2.1 POTENTIAL OUTCOMES AND THE BAYESIAN IMPUTATION MODEL

The concept of potential outcomes was proposed in Neyman [1923] for randomized trial experiments. Rubin [1975, 1976, 1977, 1978] re-formalized the framework for observational studies. We consider the causal effects of a binary treatment $w$, with $w = 1$ indicating assignment to 'treatment' and $w = 0$ indicating assignment to 'control'. Following the literature, the causal effect for individual $i$ is defined as a comparison of the two potential outcomes, $y_i(0)$ and $y_i(1)$, where these are the outcomes that would be observed under $w_i = 0$ and $w_i = 1$, respectively. We can never observe both $y_i(0)$ and $y_i(1)$ for any individual $i$, because it is not possible to go back in time and expose the $i$–th individual to the other treatment. In this work, we generalize the Bayesian imputation model of Imbens and Rubin [2015] since it captures uncertainty of the causal estimands in a Bayesian setting:

$$y_i(0) = \boldsymbol{\beta}_0^\top \mathbf{x}_i + \epsilon_{0i}, \qquad y_i(1) = \boldsymbol{\beta}_1^\top \mathbf{x}_i + \epsilon_{1i}, \qquad (1)$$

where $\epsilon_{0i}$ and $\epsilon_{1i}$ are the Gaussian noises. The key to compute treatment effects is $y_i(0)$ and $y_i(1)$. So we need to impute one of the two outcomes. Let $y_{i,\text{obs}}$, $y_{i,\text{mis}}$ be the observed and unobserved (or missing) outcome. The idea is to find the marginal distribution $p(y_{i,\text{mis}}|\mathbf{y}_{\text{obs}}, \mathbf{X}, \mathbf{w})$. Once the missing outcomes are imputed, the treatment effects can be estimated. To proceed, Imbens and Rubin [2015] suggested four steps based on the following equation $p(y_{i,\text{mis}}|\mathbf{y}_{\text{obs}}, \mathbf{X}, \mathbf{w}) =$

$\int p(y_{i,\mathrm{mis}}|\mathbf{y}_{\mathrm{obs}}, \mathbf{X}, \mathbf{w}, \theta)p(\theta|\mathbf{y}_{\mathrm{obs}}, \mathbf{X}, \mathbf{w})d\theta$, where $\theta = \{\boldsymbol{\beta}_0, \boldsymbol{\beta}_1\}$. The aim is to find $p(y_{i,\mathrm{mis}}|\mathbf{y}_{\mathrm{obs}}, \mathbf{X}, \mathbf{w}, \theta)$ and $p(\theta|\mathbf{y}_{\mathrm{obs}}, \mathbf{X}, \mathbf{w})$, and then compute the integral to obtain $p(y_{i,\mathrm{mis}}|\mathbf{y}_{\mathrm{obs}}, \mathbf{X}, \mathbf{w})$, which is a non-parametric prediction.

*The above procedure shows that learning the distribution $p(y_{i,\mathrm{mis}}|\mathbf{y}_{\mathrm{obs}}, \mathbf{X}, \mathbf{w})$ would require data from all sources since it is conditional on $\mathbf{y}_{\mathrm{obs}}$, $\mathbf{X}$, and $\mathbf{w}$. Thus, it violates the data privacy constraint.* In Sections 3.3, 3.4 and 3.5, we generalize this model with Gaussian processes and decompose it into multiple components to perform federated inference of the causal effects, which minimizes the risk of privacy leak of the data.

# 3 OUR APPROACH

We generalize the Bayesian imputation model presented in Section 2.1 to a generic model based on Gaussian Processes (GPs). We introduce the Federated Causal Inference (FedCI) method to decompose the model into multiple components, each associated with a data source, to estimate causal effects under a federated setting.

## 3.1 PROBLEM FORMULATION

**Problem setting & notations.** Suppose we have $m$ data sources that are organized and curated at their local sites. Each source is denoted by $\mathsf{D}^{\mathsf{s}} = \{(w_i^{\mathsf{s}}, y_{i,\mathrm{obs}}^{\mathsf{s}}, \mathbf{x}_i^{\mathsf{s}})\}_{i=1}^{n_{\mathsf{s}}}$, where $\mathsf{s} = 1, 2, \ldots, m$, and the quantities $w_i^{\mathsf{s}}$, $y_{i,\mathrm{obs}}^{\mathsf{s}}$ and $\mathbf{x}_i^{\mathsf{s}}$ are the treatment assignment, observed outcome associated with the treatment, and covariates of individual $i$ in source $\mathsf{s}$, respectively. In this work, we focus on binary treatment $w_i^{\mathsf{s}} \in \{0, 1\}$, thus $y_{i,\mathrm{obs}}^{\mathsf{s}}$ can be either of the potential outcomes $y_i^{\mathsf{s}}(0)$ or $y_i^{\mathsf{s}}(1)$, i.e., for each individual $i$, we can only observe either $y_i^{\mathsf{s}}(0)$ or $y_i^{\mathsf{s}}(1)$, but not both of them. We further denote the unobserved or missing outcome as $y_{i,\mathrm{mis}}^{\mathsf{s}}$. These variables are related to each other through the following equations:

$$y_i^{\mathsf{s}}(1) = w_i^{\mathsf{s}} y_{i,\mathrm{obs}}^{\mathsf{s}} + (1 - w_i^{\mathsf{s}}) y_{i,\mathrm{mis}}^{\mathsf{s}}, \quad (2)$$

$$y_i^{\mathsf{s}}(0) = (1 - w_i^{\mathsf{s}}) y_{i,\mathrm{obs}}^{\mathsf{s}} + w_i^{\mathsf{s}} y_{i,\mathrm{mis}}^{\mathsf{s}}. \quad (3)$$

Thus, $y_i^{\mathsf{s}}(1) = y_{i,\mathrm{obs}}^{\mathsf{s}}$ when $w_i^{\mathsf{s}} = 1$ and $y_i^{\mathsf{s}}(1) = y_{i,\mathrm{mis}}^{\mathsf{s}}$ when $w_i^{\mathsf{s}} = 0$, and similarly for $y_i^{\mathsf{s}}(0)$. For notational convenience, we further denote $\mathbf{y}^{\mathsf{s}}(0) = [y_1^{\mathsf{s}}(0), ..., y_{n_{\mathsf{s}}}^{\mathsf{s}}(0)]^{\top}$, $\mathbf{y}_{\mathrm{obs}}^{\mathsf{s}} = [y_{1,\mathrm{obs}}^{\mathsf{s}}, ..., y_{n_{\mathsf{s}},\mathrm{obs}}^{\mathsf{s}}]^{\top}$, and similarly for $\mathbf{y}^{\mathsf{s}}(1)$, $\mathbf{y}_{\mathrm{mis}}^{\mathsf{s}}$, $\mathbf{X}^{\mathsf{s}}$ and $\mathbf{w}^{\mathsf{s}}$.

**Causal effects of interest.** We estimate the individual treatment effect (ITE)[2] and the average treatment effect (ATE) defined as follows:

$$\tau_i^{\mathsf{s}} := y_i^{\mathsf{s}}(1) - y_i^{\mathsf{s}}(0), \quad \tau := \sum_{\mathsf{s}=1}^{m} \sum_{i=1}^{n_{\mathsf{s}}} \tau_i^{\mathsf{s}}/n, \quad (4)$$

---

[2]Also known as conditional average treatment effect (CATE).

where $y_i^{\mathsf{s}}(1)$, $y_i^{\mathsf{s}}(0)$ are realization outcomes of the corresponding random variables, and $n = \sum_{\mathsf{s}=1}^{m} n_{\mathsf{s}}$.

**The causal estimands.** Inserting Eq. (2) and (3) into (4), we obtain the estimate of ITE:

$$\mathbb{E}[\tau_i^{\mathsf{s}}] = (2w_i^{\mathsf{s}} - 1)(y_{i,\mathrm{obs}}^{\mathsf{s}} - \mathbb{E}[y_{i,\mathrm{mis}}^{\mathsf{s}}|\mathbf{y}_{\mathrm{obs}}, \mathbf{X}, \mathbf{w}]), \quad (5)$$

$$\mathbb{V}\mathrm{ar}[\tau_i^{\mathsf{s}}] = (2w_i^{\mathsf{s}} - 1)^2 \mathbb{V}\mathrm{ar}[y_{i,\mathrm{mis}}^{\mathsf{s}}|\mathbf{y}_{\mathrm{obs}}, \mathbf{X}, \mathbf{w}], \quad (6)$$

where $\mathbf{y}_{\mathrm{obs}}$, $\mathbf{X}$, $\mathbf{w}$ denote the vectors/matrices of the observed outcomes, covariates and treatments concatenated from all the sources. The estimate of ATE is as follows:

$$\mathbb{E}[\tau] = (2\mathbf{w} - \mathbf{1})^{\top}(\mathbf{y}_{\mathrm{obs}} - \mathbb{E}[\mathbf{y}_{\mathrm{mis}}|\mathbf{y}_{\mathrm{obs}}, \mathbf{X}, \mathbf{w}])/n, \quad (7)$$

$$\mathbb{V}\mathrm{ar}[\tau] = (2\mathbf{w} - \mathbf{1})^{\top}\mathbb{C}\mathrm{ov}[\mathbf{y}_{\mathrm{mis}}|\mathbf{y}_{\mathrm{obs}}, \mathbf{X}, \mathbf{w}](2\mathbf{w} - \mathbf{1})/n^2, \quad (8)$$

where $\mathbf{1}$ is a vector of ones.

Hence, the remaining task is to learn the posterior $p(\mathbf{y}_{\mathrm{mis}}|\mathbf{y}_{\mathrm{obs}}, \mathbf{X}, \mathbf{w})$, which is the predictive distribution of $\mathbf{y}_{\mathrm{mis}}$ given *all* the covariates, treatments and observed outcomes from *all* sources.

## 3.2 ASSUMPTIONS

The following assumptions are made to enable federated causal estimations:

**Assumption 1** (Strong Ignorability). *(i)* The potential outcomes are independent of the treatment assignment conditional on the covariates *(unconfoundedness)*, i.e., $y_i^{\mathsf{s}}(1), y_i^{\mathsf{s}}(0) \perp\!\!\!\perp w_i^{\mathsf{s}}|\mathbf{x}_i^{\mathsf{s}}$, and *(ii)* every individual has some positive probability to be assigned to every treatment *(positivity)*, i.e., $0 < p(w_i^{\mathsf{s}} = 1|\mathbf{x}_i^{\mathsf{s}}) < 1$. [Rosenbaum and Rubin, 1983]

**Assumption 2** (Stable Unit Treatment Value Assumption or SUTVA). *(i)* The potential outcomes for any individual do not vary with the treatments assigned to other individuals, and *(ii)* there are no different forms or versions of each treatment level, which would lead to different potential outcomes. [Imbens and Rubin, 2015]

**Assumption 3.** The individuals from all sources share the same set of common covariates.

**Assumption 4.** There exists a set of features such that any individual is uniquely identified across different sources. We refer to this set as 'primary key'.

**Assumption 5.** Data in different sources are drawn from parts of the population. The multi-source data, which may be homogeneous or heterogeneous in nature, together reflect the characteristics of the population.

Assumption 1 and 2 are standards in causal inference, as discussed in, e.g., Imbens and Rubin [2015], Shalit et al. [2017]. Assumption 3 is reasonable, e.g., decentralized data in Choudhury et al. [2019], Vaid et al. [2020], Flores et al. [2021] (to name a few) satisfy this assumption for federated learning. In Assumption 4, a 'primary key' is not limited

to the observed data used for inference as described in Section 3.1, but it can include any features to uniquely identify an individual, such as {nationality, national id} of a patient. Assumption 4 allows a secure preprocessing procedure to remove repeated individual records in different sources, if necessary, without sharing raw data among the sources (see Appendix A for details). Assumption 5 ensures that there is no imbalanced data bias across the sources. In the subsequent sections, we assume that all of the above assumptions are satisfied, and the preprocessing procedure is already performed if necessary.

## 3.3 GP-BASED IMPUTATION

The model presented in Eq. (1) is a simple Bayesian linear model. In this section, we present a more general nonlinear Bayesian model. In particular, since $\boldsymbol{\beta}_0$ and $\boldsymbol{\beta}_1$ in Eq. (1) follow multivariate normal distributions, the two components $\boldsymbol{\beta}_0^\top \mathbf{x}_i$ and $\boldsymbol{\beta}_1^\top \mathbf{x}_i$ also follow multivariate normal distributions. The generalisation of these two components are $f_0(\mathbf{x}_i) = \boldsymbol{\beta}_0^\top \omega(\mathbf{x}_i)$ and $f_1(\mathbf{x}_i) = \boldsymbol{\beta}_1^\top \omega(\mathbf{x}_i)$, where $\omega(\mathbf{x}_i)$ is a vector of basis functions with input $\mathbf{x}_i$. This formulation would lead to the fact that the marginal of $f_0(\mathbf{x})$ and $f_1(\mathbf{x})$ are Gaussian processes (GPs). Thus, we propose:

$$y_i(0) = f_0(\mathbf{x}_i) + \epsilon_{0i}, \qquad y_i(1) = f_1(\mathbf{x}_i) + \epsilon_{1i}, \qquad (9)$$

where $f_0(\mathbf{x}_i)$ and $f_1(\mathbf{x}_i)$ are two random functions evaluated at $\mathbf{x}_i$, i.e., $f_0(\mathbf{x}_i) \sim \mathsf{GP}(\mu_0(\mathbf{X}), \mathbf{K})$ and $f_1(\mathbf{x}_i) \sim \mathsf{GP}(\mu_1(\mathbf{X}), \mathbf{K})$, where $\mathbf{K}$ denotes the covariance matrix computed with a kernel function $\mathsf{k}(\mathbf{x}, \mathbf{x}')$. Similar to the imputation model of Imbens and Rubin [2015], this model also requires finding the marginal distribution $p(y_{i,\mathrm{mis}} \,|\, \mathbf{y}_{\mathrm{obs}}, \mathbf{X}, \mathbf{w})$, *through accessing the observed data from all the sources.*

*Similarly, although this model is generic, it requires access to all the observed data to compute* $\mathbf{K}$*, which is impossible without violating the privacy constraints mentioned above.* In the subsequent sections, we propose a federated model to address this problem.

## 3.4 THE PROPOSED MODEL

Recall that the aim is to find $p(\mathbf{y}_{\mathrm{mis}} \,|\, \mathbf{y}_{\mathrm{obs}}, \mathbf{X}, \mathbf{w})$ so that we may in turn compute Eqs. (5)-(8) to arrive at the quantities of interest. To that end, we propose to model the joint distribution of the potential outcomes as follows:

$$\begin{bmatrix} y_i^{\mathsf{s}}(0) \\ y_i^{\mathsf{s}}(1) \end{bmatrix} = \Psi^{\frac{1}{2}} \left( \begin{bmatrix} f_0^{\mathsf{s}}(\mathbf{x}_i) \\ f_1^{\mathsf{s}}(\mathbf{x}_i) \end{bmatrix} + \begin{bmatrix} g_0^{\mathsf{s}} \\ g_1^{\mathsf{s}} \end{bmatrix} \right) + \Sigma^{\frac{1}{2}} \, \boldsymbol{\varepsilon}_i^{\mathsf{s}}, \qquad (10)$$

where $\boldsymbol{\varepsilon}_i^{\mathsf{s}} \sim \mathsf{N}(\mathbf{0}, \mathbf{I}_2)$ is to handle the noise of the outcomes.

As mentioned in Section 2.1 and 3.3, all the outcomes from all sources are *interdependent* in the Bayesian imputation approach, which is problematic for federated learning. This dependency is handled via $f_j^{\mathsf{s}}(\mathbf{x}_i)$ and $g_j^{\mathsf{s}}$ ($j \in \{0, 1\}$), which

enable federated learning for the proposed model. We refer to the dependency handled by $f_j^{\mathsf{s}}(\mathbf{x}_i)$ as intra-dependency and the one captured by $g_j^{\mathsf{s}}$ as inter-dependency.

**Intra-dependency.** $f_0^{\mathsf{s}}(\mathbf{x}_i)$ and $f_1^{\mathsf{s}}(\mathbf{x}_i)$ are GP-distributed functions, which allows us to model each source dataset simultaneously along with its heterogeneous correlations. Specifically, we model $f_0^{\mathsf{s}}(\mathbf{x}_i) \sim \mathsf{GP}(\mu_0(\mathbf{X}^{\mathsf{s}}), \mathbf{K}^{\mathsf{s}})$ and $f_1^{\mathsf{s}}(\mathbf{x}_i) \sim \mathsf{GP}(\mu_1(\mathbf{X}^{\mathsf{s}}), \mathbf{K}^{\mathsf{s}})$, where $\mathbf{K}^{\mathsf{s}}$ is a covariance matrix computed by a kernel function $\mathsf{k}(\mathbf{x}_i^{\mathsf{s}}, \mathbf{x}_j^{\mathsf{s}})$, and $\mu_0(\cdot)$, $\mu_1(\cdot)$ are functions modelling the mean of these GPs. Parameters of these functions and hyperparameters in the kernel function are shared across multiple sources. The above GPs handle the correlations within one source only.

**Inter-dependency.** To capture *dependency* among the sources, we introduce variable $\mathbf{g} = [\mathbf{g}_0, \mathbf{g}_1]$, where $\mathbf{g}_0 = [g_0^1, ..., g_0^m]^\top \sim \mathsf{N}(\boldsymbol{r}_0, \mathbf{M})$ and $\mathbf{g}_1 = [g_1^1, ..., g_1^m]^\top \sim \mathsf{N}(\boldsymbol{r}_1, \mathbf{M})$. Both $g_0^{\mathsf{s}}$ and $g_1^{\mathsf{s}}$ are shared within the source s, and they are correlated across multiple sources $\mathsf{s} \in \{1, ..., m\}$. The correlations among the sources are modelled via the covariance matrix $\mathbf{M}$ which is computed with a kernel function. The inputs to the kernel function are the sufficient statistics (we used mean, variance, skewness, and kurtosis) of each covariate $\mathbf{x}^{\mathsf{s}}$ within the source s. We denote the first four moments of covariates as $\widetilde{\mathbf{x}}^{\mathsf{s}} \in \mathbb{R}^{4d_x \times 1}$ and the kernel function as $\gamma(\widetilde{\mathbf{x}}^{\mathsf{s}}, \widetilde{\mathbf{x}}^{\mathsf{s}'})$, which evaluates the correlation of two sources s and s′. This formulation implies that $\mathbf{g}_0$ and $\mathbf{g}_1$ are GPs. The elements of $\boldsymbol{r}_0$ and $\boldsymbol{r}_1$ are computed with the mean functions $r_0(\widetilde{\mathbf{x}}^{\mathsf{s}})$ and $r_1(\widetilde{\mathbf{x}}^{\mathsf{s}})$, respectively. Herein, we only share the sufficient statistics of covariates, but not covariates of a specific individual.

**The two variables $\Psi$ and $\Sigma$.** These are positive semi-definite matrices capturing the correlations between the two possible outcomes $y_i^{\mathsf{s}}(0)$ and $y_i^{\mathsf{s}}(1)$, $\Psi^{\frac{1}{2}}$ and $\Sigma^{\frac{1}{2}}$ are their Cholesky decomposition matrices. Note that $\Psi$ and $\Sigma$ are also random variables. Since these are positive semi-definite matrices, we model their priors using Wishart distribution $\Psi \sim \mathsf{Wishart}(\mathbf{V}_0, d_0)$, $\Sigma \sim \mathsf{Wishart}(\mathbf{S}_0, n_0)$, where $\mathbf{V}_0, \mathbf{S}_0 \in \mathbb{R}^{2 \times 2}$ are predefined positive semi-definite matrices and $d_0, n_0 \geq 2$ are predefined degrees of freedom.

## 3.5 THE PROPOSED ALGORITHM

Based on some results on the joint distribution of potential outcomes, we construct a federated objective function for the proposed federated causal inference algorithm (FedCI).

### 3.5.1 Joint Distribution of the Outcomes

We first present some results that are helpful in constructing the federated objective function in Section 3.5.2. The proofs of these results are in the appendices. To simplify the exposition, we denote $\mathbf{g}^{\mathsf{s}} = [\mathbf{g}_0^{\mathsf{s}}, \mathbf{g}_1^{\mathsf{s}}]$, where $\mathbf{g}_0^{\mathsf{s}} = [g_0^{\mathsf{s}}, ..., g_0^{\mathsf{s}}]^\top$ and $\mathbf{g}_1^{\mathsf{s}} = [g_1^{\mathsf{s}}, ..., g_1^{\mathsf{s}}]^\top$.

**Lemma 1.** *Let $\Psi$, $\Sigma$, $\mathbf{K}$, $\mu_0(\mathbf{X}^{\mathsf{s}})$, $\mu_1(\mathbf{X}^{\mathsf{s}})$, and $\mathbf{g}^{\mathsf{s}}$ satisfy the model in Eq.* (10). *Then,*

$$
\begin{bmatrix} \mathbf{y}^{\mathsf{s}}(0) \\ \mathbf{y}^{\mathsf{s}}(1) \end{bmatrix} \Big| \Psi, \Sigma, \mathbf{X}^{\mathsf{s}}, \mathbf{w}^{\mathsf{s}}, \mathbf{g}^{\mathsf{s}}
$$
$$
\sim \mathsf{N}\left( \left(\Psi^{\frac{1}{2}} \otimes \mathbf{I}_{n_{\mathsf{s}}}\right) \begin{bmatrix} \mu_0(\mathbf{X}^{\mathsf{s}}) + \mathbf{g}_0^{\mathsf{s}} \\ \mu_1(\mathbf{X}^{\mathsf{s}}) + \mathbf{g}_1^{\mathsf{s}} \end{bmatrix}, \Psi \otimes \mathbf{K}^{\mathsf{s}} + \Sigma \otimes \mathbf{I}_{n_{\mathsf{s}}} \right),
$$

*where $\otimes$ is the Kronecker product.*

The proof of Lemma 1 is presented in Appendix C. From Lemma 1, we observe that $\Psi$, $\mathbf{K}^{\mathsf{s}}$, $\Sigma$, and $\mathbf{I}_{n_{\mathsf{s}}}$ are positive semi-definite, thus the covariance matrix $\Psi \otimes \mathbf{K}^{\mathsf{s}} + \Sigma \otimes \mathbf{I}_{n_{\mathsf{s}}}$ is positive semi-definite due to the fundamental property of Kronecker product. This is why we choose $\Psi$ and $\Sigma$ to be positive semi-definite in our model; otherwise, the covariance matrix is invalid. From Lemma 1, we can obtain the result in Lemma 2 as follows:

**Lemma 2.** *Let $\Psi$, $\Sigma$, $\mathbf{K}$, $\mu_0(\mathbf{X}^{\mathsf{s}})$, $\mu_1(\mathbf{X}^{\mathsf{s}})$, and $\mathbf{g}^{\mathsf{s}}$ satisfy the model in Eq.* (10). *Then,*

$$
\begin{bmatrix} \mathbf{y}_{\mathrm{obs}}^{\mathsf{s}} \\ \mathbf{y}_{\mathrm{mis}}^{\mathsf{s}} \end{bmatrix} \Big| \Psi, \Sigma, \mathbf{X}^{\mathsf{s}}, \mathbf{w}^{\mathsf{s}}, \mathbf{g}^{\mathsf{s}} \sim \mathsf{N}\left( \begin{bmatrix} \mu_{\mathrm{obs}}(\mathbf{X}^{\mathsf{s}}) \\ \mu_{\mathrm{mis}}(\mathbf{X}^{\mathsf{s}}) \end{bmatrix}, \begin{bmatrix} \mathbf{K}_{\mathrm{obs}}^{\mathsf{s}} & \mathbf{K}_{\mathrm{om}}^{\mathsf{s}} \\ (\mathbf{K}_{\mathrm{om}}^{\mathsf{s}})^{\top} & \mathbf{K}_{\mathrm{mis}}^{\mathsf{s}} \end{bmatrix} \right).
$$

*The mean functions $\mu_{\mathrm{obs}}(\mathbf{X}^{\mathsf{s}})$ and $\mu_{\mathrm{mis}}(\mathbf{X}^{\mathsf{s}})$ are:*

$$
\mu_{\mathrm{obs}}(\mathbf{X}^{\mathsf{s}}) = (\mathbf{1} - \mathbf{w}^{\mathsf{s}}) \odot \mathbf{m}_0 + \mathbf{w}^{\mathsf{s}} \odot \mathbf{m}_1,
$$
$$
\mu_{\mathrm{mis}}(\mathbf{X}^{\mathsf{s}}) = \mathbf{w}^{\mathsf{s}} \odot \mathbf{m}_0 + (\mathbf{1} - \mathbf{w}^{\mathsf{s}}) \odot \mathbf{m}_1,
$$

*where we denote $\mathbf{m}_0 = \psi_{11}^*(\mu_0(\mathbf{X}^{\mathsf{s}}) + \mathbf{g}_0^{\mathsf{s}})$ and $\mathbf{m}_1 = \psi_{21}^*(\mu_0(\mathbf{X}^{\mathsf{s}}) + \mathbf{g}_0^{\mathsf{s}}) + \psi_{22}^*(\mu_1(\mathbf{X}^{\mathsf{s}}) + \mathbf{g}_1^{\mathsf{s}})$ with $\psi_{ij}^*$ is the $(i,j)$–th element of Cholesky decomposition matrix of $\Psi$, $\mathbf{1}$ is a vector ones, and $\odot$ is the element-wise product. The covariance matrices $\mathbf{K}_{\mathrm{obs}}^{\mathsf{s}}$, $\mathbf{K}_{\mathrm{mis}}^{\mathsf{s}}$, and $\mathbf{K}_{\mathrm{om}}^{\mathsf{s}}$ are computed by kernel functions:*

$$
\begin{aligned}
k_{\mathrm{obs}}(\mathbf{x}_i, \mathbf{x}_j) = & \big[(1-w_i)(1-w_j)\psi_{11} + w_i w_j \psi_{22} \\
& + (1-w_i)w_j \psi_{12} + w_i(1-w_j)\psi_{21}\big] \mathsf{k}(\mathbf{x}_i, \mathbf{x}_j) \\
& + \big[(1-w_i)\sigma_{11} + w_i \sigma_{22}\big] \mathbb{1}_{i=j},
\end{aligned}
$$

$$
\begin{aligned}
k_{\mathrm{mis}}(\mathbf{x}_i, \mathbf{x}_j) = & \big[w_i w_j \psi_{11} + (1-w_i)(1-w_j)\psi_{22} \\
& + (1-w_i)w_j \psi_{21} + w_i(1-w_j)\psi_{12}\big] \mathsf{k}(\mathbf{x}_i, \mathbf{x}_j) \\
& + \big[w_i \sigma_{11} + (1-w_i)\sigma_{22}\big] \mathbb{1}_{i=j},
\end{aligned}
$$

$$
\begin{aligned}
k_{\mathrm{om}}(\mathbf{x}_i, \mathbf{x}_j) = & \big[(1-w_i)(1-w_j)\psi_{21} + w_i w_j \psi_{12} \\
& + (1-w_i)w_j \psi_{22} + w_i(1-w_j)\psi_{11}\big] \mathsf{k}(\mathbf{x}_i, \mathbf{x}_j) \\
& + \big[(1-w_i)\sigma_{21} + w_i \sigma_{12}\big] \mathbb{1}_{i=j},
\end{aligned}
$$

*where $\psi_{ab}$ and $\sigma_{ab}$ are the $(a,b)$–th elements of $\Psi$ and $\Sigma$, respectively.*

The proof of Lemma 2 is in Appendix D. Lemma 2 has two important roles in our work. First, we can obtain the conditional likelihood to help infer the parameters and hyperparameters of our proposed model. Second, we can also obtain the posterior of $\mathbf{y}_{\mathrm{mis}}^{\mathsf{s}}$ to help us estimate ITE and ATE.

### 3.5.2 Federated Objective Function

The proposed model in Eq. (10) would lead to an objective function that can be decomposed into $m$ components, each associated with a data source. Since estimating $p(\mathbf{y}_{\mathrm{mis}}^{\mathsf{s}} \mid \mathbf{y}_{\mathrm{obs}}^{\mathsf{s}}, \mathbf{X}^{\mathsf{s}}, \mathbf{w}^{\mathsf{s}})$ exactly is intractable, we sidestep this intractability via a variational approximation. To achieve this, we maximize the following evidence lower bound (ELBO) $\mathbf{L}$:

$$
\log p(\mathbf{y}_{\mathrm{obs}} \mid \mathbf{X}, \mathbf{w}) = \log \int p(\mathbf{y}_{\mathrm{obs}}, \mathbf{g}, \Psi, \Sigma \mid \mathbf{X}, \mathbf{w}) d\mathbf{g} d\Psi d\Sigma
$$
$$
\geq \sum_{\mathsf{s}=1}^{m} \mathbf{L}^{\mathsf{s}} =: \mathbf{L}, \tag{11}
$$

where

$$
\mathbf{L}^{\mathsf{s}} = \mathbb{E}_q\left[\log p(\mathbf{y}_{\mathrm{obs}}^{\mathsf{s}} | \cdot)\right] - \frac{1}{m}\left( \sum_{z \in \{\mathbf{g}, \Psi, \Sigma\}} \mathbb{D}_{\mathrm{KL}}[q(z)\|p(z)] \right).
$$

Herein, $\mathbb{D}_{\mathrm{KL}}[\cdot]$ is the Kullback–Leibler divergence. Details of the ELBO are presented in Appendix B. The conditional likelihood $p(\mathbf{y}_{\mathrm{obs}}^{\mathsf{s}} | \cdot)$ is obtained from Lemma 2 by marginalizing out $\mathbf{y}_{\mathrm{mis}}^{\mathsf{s}}$, i.e.,

$$
p(\mathbf{y}_{\mathrm{obs}}^{\mathsf{s}} | \mathbf{X}^{\mathsf{s}}, \mathbf{w}^{\mathsf{s}}, \Psi, \Sigma, \mathbf{g}^{\mathsf{s}}) = \mathsf{N}(\mathbf{y}_{\mathrm{obs}}^{\mathsf{s}}; \mu_{\mathrm{obs}}(\mathbf{X}^{\mathsf{s}}), \mathbf{K}_{\mathrm{obs}}^{\mathsf{s}}). \tag{12}
$$

The above conditional likelihood is free of $\sigma_{21}$ and $\sigma_{12}$, which capture the correlation of two potential outcomes. Thus the posteriors of these variables would coincide with their priors, i.e., the correlation cannot be learned but set as a prior. This is well-known as one of the potential outcome cannot be observed [Imbens and Rubin, 2015]. In Eq. (11), the ELBO $\mathbf{L}$ is derived from the of joint marginal likelihood of all $m$ sources, and it is factorized into $m$ components $\mathbf{L}^{\mathsf{s}}$, each component corresponds to a source. This enables federated optimization of $\mathbf{L}$. The first term of $\mathbf{L}^{\mathsf{s}}$ is expectation of the conditional likelihood with respect to the variational posterior $q(\mathbf{g}, \Psi, \Sigma)$, thus this distribution is learned from data of all the sources. In the following, we present its factorization.

**Variational posterior distributions.** We use the typical mean-field approximation to factorize among the variational posteriors $q(\Psi, \Sigma, \mathbf{g}) = q(\Psi)\, q(\Sigma)\, q(\mathbf{g})$. Let $\widetilde{\mathbf{y}}_{\mathrm{obs}}^{\mathsf{s}}(0)$, $\widetilde{\mathbf{y}}_{\mathrm{obs}}^{\mathsf{s}}(1)$, $\widetilde{\mathbf{x}}^{\mathsf{s}}$, and $\widetilde{\mathbf{w}}^{\mathsf{s}}$ ($\mathsf{s} = 1, 2, ..., m$) be the first four moments of the observed outcomes, covariates, and treatment of the $\mathsf{s}$–th source. Let $\widetilde{\mathbf{X}} = [\widetilde{\mathbf{x}}^1, ..., \widetilde{\mathbf{x}}^m]^{\top}$, $\widetilde{\mathbf{y}}_{\mathrm{obs}}(0) = [\widetilde{\mathbf{y}}_{\mathrm{obs}}^1(0), ..., \widetilde{\mathbf{y}}_{\mathrm{obs}}^m(0)]^{\top}$, $\widetilde{\mathbf{y}}_{\mathrm{obs}}(1) = [\widetilde{\mathbf{y}}_{\mathrm{obs}}^1(1), ..., \widetilde{\mathbf{y}}_{\mathrm{obs}}^m(1)]^{\top}$, and $\widetilde{\mathbf{w}} = [\widetilde{\mathbf{w}}^1, ..., \widetilde{\mathbf{w}}^m]^{\top}$. We parameterize

$$
q(\mathbf{g}) = \prod_{j \in \{0,1\}} \mathsf{N}(\mathbf{g}_j; h_j(\widetilde{\mathbf{y}}_{\mathrm{obs}}(0), \widetilde{\mathbf{y}}_{\mathrm{obs}}(1), \widetilde{\mathbf{X}}, \widetilde{\mathbf{w}}), \mathbf{U}),
$$

where $h_0(\cdot)$ and $h_1(\cdot)$ are the mean functions, $\mathbf{U}$ is the covariance matrix computed with a kernel function $\kappa(u^{\mathsf{s}}, u^{\mathsf{s}'})$, where $u^{\mathsf{s}} := [\widetilde{\mathbf{y}}_{\mathrm{obs}}^{\mathsf{s}}(0), \widetilde{\mathbf{y}}_{\mathrm{obs}}^{\mathsf{s}}(1), \widetilde{\mathbf{x}}^{\mathsf{s}}, \widetilde{\mathbf{w}}^{\mathsf{s}}]$.

Since $\Psi$ and $\Sigma$ are positive semi-definite matrices, we model their variational posterior as Wishart distribution:

$$q(\Psi) = \mathsf{Wishart}(\Psi; \mathbf{V}_q, d_q), \quad q(\Sigma) = \mathsf{Wishart}(\Sigma; \mathbf{S}_q, n_q),$$

where $d_q, n_q$ are degrees of freedom, $\mathbf{V}_q, \mathbf{S}_q$ are the scale matrices. We set the form of these scale matrices as follows

$$\mathbf{V}_q = \begin{bmatrix} \nu_1^2 & \rho\nu_1\nu_2 \\ \rho\nu_1\nu_2 & \nu_2^2 \end{bmatrix}, \quad \mathbf{S}_q = \begin{bmatrix} \delta_1^2 & \eta\delta_1\delta_2 \\ \eta\delta_1\delta_2 & \delta_2^2 \end{bmatrix},$$

where $\nu_i, \rho, \delta_i, \eta$ are parameters to be learned and $\rho, \eta \in [0, 1]$.

**Reparameterization.** To maximize the ELBO, we approximate the expectation in $\mathbf{L}^{\mathsf{s}}$ with Monte Carlo integration, which requires drawing samples of $\mathbf{g}, \Psi$ and $\Sigma$ from their variational distributions. This requires a reparameterization to allow the gradients to pass through the random variables $\mathbf{g}, \Psi$ and $\Sigma$. The reparameterization trick for $\mathbf{g}$ are: $\mathbf{g}_j = h_j(\widetilde{\mathbf{y}}_{\mathrm{obs}}(0), \widetilde{\mathbf{y}}_{\mathrm{obs}}(1), \widetilde{\mathbf{X}}, \widetilde{\mathbf{w}}) + \mathbf{U}^{\frac{1}{2}}\boldsymbol{\xi}_j, j \in \{0, 1\}$, where $\boldsymbol{\xi}_j \sim \mathsf{N}(\mathbf{0}, \mathbf{I}_m)$ and $\mathbf{U}^{\frac{1}{2}}$ is the Cholesky decomposition matrix of $\mathbf{U}$. Since $q(\Psi)$ is a Wishart distribution, we introduce the following procedure to draw $\Psi$: $\Psi = \mathbf{V}_q^{\frac{1}{2}}\boldsymbol{\zeta}(\mathbf{V}_q^{\frac{1}{2}})^\top, \boldsymbol{\zeta} \sim \mathsf{Wishart}(\mathbf{I}_2, d_q)$, where $\mathbf{V}_q^{\frac{1}{2}}$ is the Choleskey decomposition matrix of $\mathbf{V}_q$. Likewise, we also apply this procedure to draw $\Sigma$.

**Federated optimization algorithm.** With the above model and its objective function, we can compute gradients of the learnable parameters separately in each source without sharing data to a central server. We summarize our procedure in Algorithm 1.

### 3.5.3 Predicting Causal Effects from Multiple Sources

To understand why data from all the sources can help predict causal effects in a source $\mathsf{s}$, we observe that

$$\begin{aligned} p(\mathbf{y}_{\mathrm{mis}}^{\mathsf{s}} \,|\, \mathbf{y}_{\mathrm{obs}}, \mathbf{X}, \mathbf{w}) \quad &(13) \\ \simeq \mathbb{E}_q\big[p(\mathbf{y}_{\mathrm{mis}}^{\mathsf{s}} | \mathbf{y}_{\mathrm{obs}}^{\mathsf{s}}, \mathbf{X}^{\mathsf{s}}, \mathbf{w}^{\mathsf{s}}, \Psi, \Sigma, \mathbf{g})\big] \\ = p(\mathbf{y}_{\mathrm{mis}}^{\mathsf{s}} \,|\, \underbrace{\mathbf{y}_{\mathrm{obs}}^{\mathsf{s}}, \mathbf{X}^{\mathsf{s}}, \mathbf{w}^{\mathsf{s}}}_{\textbf{(i)}}, \underbrace{\Theta}_{\textbf{(ii)}}, \underbrace{\widetilde{\mathbf{y}}_{\mathrm{obs}}(0), \widetilde{\mathbf{y}}_{\mathrm{obs}}(1), \widetilde{\mathbf{X}}, \widetilde{\mathbf{w}}}_{\textbf{(iii)}}). \end{aligned}$$

Eq. (13) is an approximation of the predictive distribution of the missing outcomes $\mathbf{y}_{\mathrm{mis}}^{\mathsf{s}}$ and it depends on the following three components:

**(i).** The observed outcomes, covariates and treatment assignments from the same source $\mathsf{s}$.

**(ii).** The shared parameters $\Theta$ learned from data of all the sources.

**(iii).** Sufficient statistics of the observed data from all the sources.

The two last components **(ii)** and **(iii)** indicate that the predictive distribution in source $\mathsf{s}$ utilizes knowledge from

all the sources through $\Theta$ and the sufficient statistics $[\widetilde{\mathbf{y}}_{\mathrm{obs}}(0), \widetilde{\mathbf{y}}_{\mathrm{obs}}(1), \widetilde{\mathbf{X}}, \widetilde{\mathbf{w}}]$. This explain why data from all of the sources help predict missing outcomes in source $\mathsf{s}$.

---

**Algorithm 1:** Federated causal inference

> **Parameters:** Let $\Theta$ be set of parameters
> 1 **begin**
> 2     Initialize $\Theta$ and send to all source machines;
> 3     **repeat**
> 4        **for** source machine $\mathsf{s} \in \{1, 2, \ldots, m\}$ **do**
> 5           Compute $\nabla_\Theta \mathbf{L}^{\mathsf{s}}$ and send to server;
> 6        In the central server, do the following steps:
> 7        **begin**
> 8           Collect gradients from all sources;
> 9           Compute $\nabla_\Theta \mathbf{L} = \sum_{\mathsf{s}=1}^m \nabla_\Theta \mathbf{L}^{\mathsf{s}}$;
> 10          Update $\Theta \leftarrow \Theta + \mathsf{learning\_rate} \times \nabla_\Theta \mathbf{L}$;
> 11          Broadcast the new $\Theta$ to all sources;
> 12     **until** stopping condition;

---

## 4 EXPERIMENTS

**The baselines and experimental objectives.** We first examine the performance of FedCI. We then compare the performance of FedCI against recent causal inference methods, such as BART [Hill, 2011], TARNet, CFR Wass (CFR-Net with Wasserstein distance), CFR MMD (CFRNet with maximum mean discrepancy distance) [Shalit et al., 2017], CEVAE [Louizos et al., 2017], OrthoRF [Oprescu et al., 2019], X-learner [Künzel et al., 2019], and R-learner [Nie and Wager, 2021]. All these methods do not consider learning causal effects in a federated setting. This analysis aims to show the efficacy of FedCI as compared with the baselines trained in three different cases: (**1**) training a local model on each source data, (**2**) training a global model with the combined data of all sources, (**3**) using bootstrap aggregating (also known as bagging, which is an ensemble learning method) of Breiman [1996] where $m$ models are locally trained on each source data; then taking average of the predicted treatment effects of each model. Although case (**2**) *violates* the privacy constraint of federated data, we use it for comparison purposes. In general, we would like to assess the performance of the federated causal inference approach against the baselines using combined data in case (**2**).

We use publicly available libraries and source codes to implement the baseline methods. In particular, CEVAE, TARNet, CFR Wass, and CFR MMD are readily available on github. We use the online packages `BartPy` for BART, `causalml` [Chen et al., 2020b] for X-learner and R-learner, and `econml` [Microsoft Research, 2019] for OrthoRF. For all the methods, we fine-tune the learning rate in $\{10^{-1}, 10^{-2}, 10^{-3}, 10^{-4}\}$ and regularizers in $\{10^1, 10^0, 10^{-1}, 10^{-2}, 10^{-3}\}$.

**Evaluation metrics.** We report two evaluation metrics: (i) precision in estimation of heterogeneous effects (PEHE) [Hill, 2011] for evaluating ITE, and (ii) absolute error for

evaluating ATE. Details are presented in Appendix E. These metrics are for point estimates, which are the mean of ITE and ATE in their estimated distributions. We also report the estimated distribution of ATE in our model.

## 4.1 SYNTHETIC DATA

We analyses FedCI in terms of three types of outcomes: (1) real-value, (2) binary, and (3) count. While (1) is examined in a well-specified case for the outcomes, (2) and (3) are studied in misspecified cases.

### 4.1.1 Real-value Outcomes

**Data.** Obtaining ground truth for evaluating causal inference algorithm is challenging. Thus, most methods are evaluated using synthetic or semi-synthetic datasets. In this experiment, we simulate the data with the following distributions:

$$x_{ij} \sim \mathsf{U}[-1, 1], \quad y_i(0) \sim \mathsf{N}(\lambda(b_0 + \mathbf{x}_i^\top \mathbf{b}_1), \sigma_0^2),$$
$$w_i \sim \mathsf{Bern}(\varphi(a_0 + \mathbf{x}_i^\top \mathbf{a}_1)), \quad y_i(1) \sim \mathsf{N}(\lambda(c_0 + \mathbf{x}_i^\top \mathbf{c}_1), \sigma_1^2),$$

where $\varphi(\cdot)$ is the sigmoid function, $\lambda(\cdot)$ is the softplus function, and $\mathbf{x}_i = [x_{i1}, ..., x_{id_x}]^\top \in \mathbb{R}^{d_x}$ with $d_x = 20$. We simulate two synthetic datasets: DATA-1 and DATA-2. For DATA-1, the ground truth parameters are randomly set as follows: $\sigma_0 = \sigma_1 = 1$, $(a_0, b_0, c_0) = (0.6, 0.9, 2.0)$, $\mathbf{a}_1 \sim \mathsf{N}(\mathbf{0}, 2 \cdot \mathbf{I}_{d_x})$, $\mathbf{b}_1 \sim \mathsf{N}(\mathbf{0}, 2 \cdot \mathbf{I}_{d_x})$, $\mathbf{c}_1 \sim \mathsf{N}(\mathbf{1}, 2 \cdot \mathbf{I}_{d_x})$. For DATA-2, we set $(b_0, c_0) = (6, 30)$, $\mathbf{b}_1 \sim \mathsf{N}(10 \cdot \mathbf{1}, 2 \cdot \mathbf{I}_{d_x})$, $\mathbf{c}_1 \sim \mathsf{N}(15 \cdot \mathbf{1}, 2 \cdot \mathbf{I}_{d_x})$, and the other parameters are set similar to that of DATA-1. The purpose is to make two different scales of the outcomes for the two datasets. For each dataset, we simulate 10 replications with $n = 5000$ records. We only keep $\{(y_i, w_i, \mathbf{x}_i)\}_{i=1}^n$ as the observed data, where $y_i = y_i(0)$ if $w_i = 0$ and $y_i = y_i(1)$ if $w_i = 1$. We divide the data into five sources, each consists of $n_s = 1000$ records. In each source, we use 50 records for training, 450 for testing and 400 for validation. We report the evaluation metrics and their standard errors over the 10 replications. The parameters chosen for this simulation study satisfy Assumption 1 since $y_i(0)$ and $y_i(1)$ are independent of $w_i$ given $\mathbf{x}_i$. Assumption 2 is respected as the treatment on an individual $i$ does not effect the outcome of another individual $j$ ($i \neq j$). Since we fixed the dimension of $\mathbf{x}_i$ and draw it from the same distribution, Assumption 3 is implicitly satisfied. Assumption 4 holds true since each record drawn from the above distributions is attributed to one individual. This means that there are no duplicates of individuals in more than one source. Assumption 5 is also satisfied since we have divided the data equally from one dataset.

**FedCI vs. training on combined data.** Figure 1 reports the three evaluation metrics of FedCI compared with two data source settings: training on combined data and training locally on each data source. As expected, the figures show that the errors of FedCI are as low as those of training on

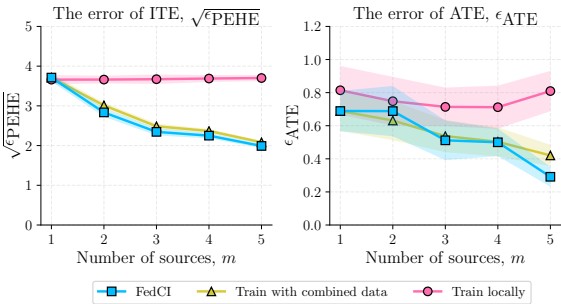

Figure 1: Federated inference analysis on DATA-1.

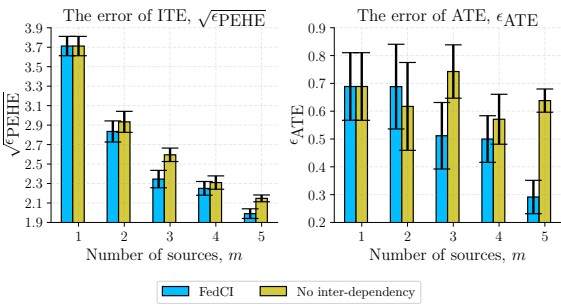

Figure 2: The impact of inter-dependency on DATA-1.

Table 1: Out-of-sample errors on DATA-1 where top-3 performances are highlighted in bold (lower is better). The dashes (——) in 'loc' and 'agg' indicate that the numbers are the same as those of 'com'.

| Method | The error of ITE ($\sqrt{\epsilon_{\text{PEHE}}}$) | | | The error of ATE ($\epsilon_{\text{ATE}}$) | | |
|---|---|---|---|---|---|---|
| | 1 source | 3 sources | 5 sources | 1 source | 3 sources | 5 sources |
| BART$_\text{loc}$ | — | 6.04±.05 | 6.02±.04 | — | 0.59±.14 | 0.53±.10 |
| X-Learner$_\text{loc}$ | — | 5.81±.13 | 5.77±.09 | — | 0.44±.24 | 0.51±.13 |
| R-Learner$_\text{loc}$ | — | 5.94±.05 | 5.94±.03 | — | 0.65±.05 | 0.66±.02 |
| OthoRF$_\text{loc}$ | — | 5.83±.12 | 6.23±.13 | — | **0.31±.08** | 0.52±.10 |
| TARNet$_\text{loc}$ | — | 4.25±.07 | 4.22±.06 | — | 0.85±.04 | 0.81±.02 |
| CFR Wass$_\text{loc}$ | — | 4.10±.04 | 3.92±.03 | — | 0.81±.02 | 0.80±.02 |
| CFR MMD$_\text{loc}$ | — | 4.11±.06 | 3.93±.03 | — | 0.80±.03 | 0.79±.02 |
| CEVAE$_\text{loc}$ | — | 3.82±.09 | 3.50±.06 | — | 0.63±.11 | 0.52±.03 |
| BART$_\text{agg}$ | — | 5.97±.05 | 5.94±.03 | — | 0.64±.14 | 0.47±.11 |
| X-Learner$_\text{agg}$ | — | 5.18±.09 | 5.09±.05 | — | 0.46±.24 | 0.52±.13 |
| R-Learner$_\text{agg}$ | — | 5.94±.05 | 5.93±.03 | — | 0.65±.05 | 0.66±.03 |
| OthoRF$_\text{agg}$ | — | 4.19±.13 | 3.66±.08 | — | **0.36±.13** | 0.48±.12 |
| TARNet$_\text{agg}$ | — | 4.02±.04 | 4.00±.05 | — | 0.79±.04 | 0.77±.02 |
| CFR Wass$_\text{agg}$ | — | 3.92±.03 | 3.75±.03 | — | 0.78±.03 | 0.76±.02 |
| CFR MMD$_\text{agg}$ | — | 4.01±.05 | 3.80±.02 | — | 0.78±.03 | 0.76±.02 |
| CEVAE$_\text{agg}$ | — | 3.65±.10 | 2.99±.06 | — | 0.41±.05 | 0.37±.04 |
| BART$_\text{com}$ | 5.98±.06 | 5.97±.06 | 5.93±.03 | 0.83±.11 | 0.56±.16 | 0.38±.09 |
| X-Learner$_\text{com}$ | 5.48±.15 | 4.60±.09 | 4.15±.04 | 0.93±.22 | 0.60±.11 | **0.30±.07** |
| R-Learner$_\text{com}$ | 5.93±.06 | 5.73±.08 | 5.54±.06 | 0.78±.10 | 0.47±.09 | **0.30±.07** |
| OthoRF$_\text{com}$ | 5.86±.40 | **3.60±.12** | **2.94±.05** | **0.55±.14** | 0.45±.14 | 0.34±.09 |
| TARNet$_\text{com}$ | 3.93±.07 | 3.87±.05 | 3.80±.03 | 0.80±.04 | 0.77±.04 | 0.76±.02 |
| CFR Wass$_\text{com}$ | **3.77±.05** | 3.73±.04 | 3.71±.02 | 0.80±.04 | 0.75±.04 | 0.75±.02 |
| CFR MMD$_\text{com}$ | 3.90±.06 | 3.73±.04 | 3.70±.02 | 0.82±.05 | 0.75±.04 | 0.75±.02 |
| CEVAE$_\text{com}$ | **3.79±.07** | **2.85±.06** | **2.72±.04** | **0.51±.13** | **0.23±.07** | **0.20±.06** |
| FedCI | **3.71±.10** | **2.35±.09** | **1.99±.05** | **0.69±.12** | **0.31±.12** | **0.29±.06** |

the combined data. This result verifies the efficacy of the proposed federated algorithm.

**Inter-dependency component analysis.** We study the impact of the inter-dependency component (see Section 3.4)

Table 2: Out-of-sample errors on DATA-2. Please see the full table in Appendix F.1.

| Method | The error of ITE ($\sqrt{\epsilon_{\text{PEHE}}}$) | | | The error of ATE ($\epsilon_{\text{ATE}}$) | | |
|---|---|---|---|---|---|---|
| | 1 source | 3 sources | 5 sources | 1 source | 3 sources | 5 sources |
| BART$_{\text{com}}$ | **18.0±0.4** | **17.7±0.2** | 17.4±0.1 | **3.54±1.3** | 2.94±0.8 | **1.84±0.5** |
| X-Learner$_{\text{com}}$ | 21.1±0.9 | 17.9±0.4 | **16.2±0.2** | 4.55±1.4 | 3.29±1.0 | 2.37±0.8 |
| R-Learner$_{\text{com}}$ | 25.9±0.6 | 23.5±0.5 | 21.3±0.4 | 19.0±0.8 | 15.6±0.7 | 12.3±0.6 |
| OthoRF$_{\text{com}}$ | 37.8±2.7 | **10.7±0.5** | **9.83±0.5** | 7.88±2.2 | **1.99±0.4** | 2.36±0.6 |
| TARNet$_{\text{com}}$ | 36.1±0.4 | 35.5±0.2 | 35.0±0.2 | 7.11±0.4 | 7.10±0.3 | 7.08±0.2 |
| CFR Wass$_{\text{com}}$ | 35.1±0.4 | 34.5±0.2 | 34.1±0.2 | 7.10±0.4 | 7.01±0.3 | 6.90±0.2 |
| CFR MMD$_{\text{com}}$ | 35.1±0.4 | 35.0±0.2 | 34.9±0.2 | 7.12±0.4 | 7.02±0.3 | 7.01±0.2 |
| CEVAE$_{\text{com}}$ | **20.1±0.5** | 18.4±0.6 | 16.6±0.6 | **1.50±0.3** | 1.38±0.4 | 1.89±0.2 |
| FedCI | **9.28±0.4** | **6.34±0.2** | **5.53±0.1** | **2.37±0.5** | **1.47±0.4** | **0.74±.2** |

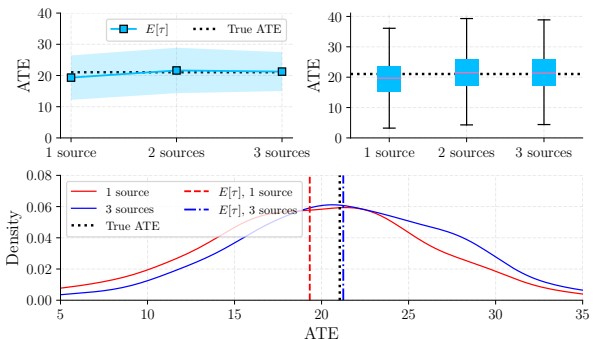

Figure 3: Estimated distribution of ATE on source #1 of DATA-2. The dotted black lines represent the true ATE.

by removing it from the model. Figure 2 presents the errors of FedCI compared with 'no inter-dependency' (FedCI without inter-dependency). The figures show that the errors in predicting ITE and ATE of 'no inter-dependency' seem to be higher than those of FedCI. This result showcases the importance of our proposed inter-dependency component.

In Figure 1, the error $\epsilon_{\text{ATE}}$ of FedCI increases as the number of sources increases from 1 to 2. In Figure 2, $\epsilon_{\text{ATE}}$ of FedCI is larger than that of without inter-dependency. These results might be due to the non-convex optimisation which could lead to a local minima. A potential direction to improve is to use a minibatch stochastic gradient descent for GPs [Chen et al., 2020a].

**Contrasting with existing baselines.** In this experiment, we compare FedCI with the existing causal inference methods. All these baseline methods do not consider estimating causal effects on multiple sources. Thus, we train them in three cases as explained earlier: (**1**) train locally (loc), (**2**) train with combined data (com), and (**3**) train with bootstrap aggregating (agg). Note that case (**2**) violates constraint that data are stored at their local sites. We expect that the error of FedCI to be close to case (**2**) of the baselines. Table 1 and 2 report the performance of each method in estimating ATE and ITE. Regardless of different scales on the two synthetic datasets, the figure shows that FedCI achieves competitive results as compared with all the baselines. FedCI is in the top-3 performances among all the methods. Importantly, FedCI obtains lower errors than those of BART$_{\text{com}}$, X-Learner$_{\text{com}}$, R-Learner$_{\text{com}}$, OthoRF$_{\text{com}}$, TARNet$_{\text{com}}$, CFR Wass$_{\text{com}}$, and CFR MMD$_{\text{com}}$, which were trained on combined data and thus violate constraint of federated data setting. Compared with CEVAE$_{\text{com}}$, FedCI is better than this method in predicting ITE and comparable with this method in predicting ATE (slightly higher errors). However, we emphasize again that this result is expected since FedCI is a federated learning algorithm while CEVAE$_{\text{com}}$ works directly on combined data.

**The estimated distribution of ATE.** To analyse uncertainty, we present in Figure 3 the estimated distribution of ATE in the first source (s = 1). The figures show that the true

ATE is covered by the estimated interval and the estimated mean ATE shifts towards its true value (dotted lines) when more data sources are used. This result might provide useful information about the application in practice.

### 4.1.2 Misspecification Analysis: Binary and Count Outcomes

**Data.** In this experiment, we analyse the performance when the model is misspecified. We compare FedCI with the baselines in two cases: binary outcomes and count outcomes. We reuse the ground truth distributions of $x_{ij}$ and $w_i$ as in Section 4.1.1. For the outcomes, we simulate them with the following distributions:

Binary outcomes:
$$y_i(0) \sim \text{Bern}(\varphi(b_0 + \mathbf{x}_i^\top \mathbf{b}_1)),$$
$$y_i(1) \sim \text{Bern}(\varphi(c_0 + \mathbf{x}_i^\top \mathbf{c}_1)).$$
Count outcomes:
$$y_i(0) \sim \text{Poisson}(\exp(b_0 + \mathbf{x}_i^\top \mathbf{b}_1)),$$
$$y_i(1) \sim \text{Poisson}(\exp(c_0 + \mathbf{x}_i^\top \mathbf{c}_1)).$$

**Results and discussion.** From Table 3 and 4, FedCI gives competitive results compared with the baselines trained on combined data. The reason for the good performance for FedCI and some baselines in these misspecification cases is because they provide good estimates for the mean of the missing outcomes. This might in turn be due to the mean estimation of Gaussian distribution in FedCI coincides with the mean estimation of the other distributions. Nevertheless, since these are misspecified cases, the continuous posterior distribution is not a good estimation. To obtain better posterior distributions of the missing outcomes and the causal estimands, we would need to consider some other appropriate distributions in our model.

### 4.2 IHDP DATA

**Data.** The Infant Health and Development Program (IHDP) [Hill, 2011] is a dataset with 747 data points, each has 25 covariates. These data are obtained from a randomized study on the impact of specialist visits to children's cognitive

Table 3: Out-of-sample errors on binary outcomes data.

| Method | The error of ITE ($\sqrt{\epsilon_{PEHE}}$) | | | The error of ATE ($\epsilon_{ATE}$) | | |
|---|---|---|---|---|---|---|
| | 1 source | 3 sources | 5 sources | 1 source | 3 sources | 5 sources |
| BART$_{com}$ | 0.77±.01 | 0.73±.01 | 0.70±.01 | 0.41±.01 | 0.31±.01 | 0.24±.01 |
| X-Learner$_{com}$ | 0.69±.01 | 0.60±.01 | 0.56±.01 | **0.13±.03** | 0.10±.02 | **0.09±.01** |
| R-Learner$_{com}$ | 0.65±.01 | 0.64±.01 | 0.62±.01 | **0.05±.01** | **0.03±.01** | **0.03±.01** |
| OthoRF$_{com}$ | 0.94±.04 | 0.60±.01 | 0.56±.01 | 0.17±.03 | 0.18±.03 | 0.16±.03 |
| TARNet$_{com}$ | 0.68±.02 | 0.68±.02 | 0.65±.01 | 0.33±.01 | 0.33±.01 | 0.32±.01 |
| CFR Wass$_{com}$ | 0.61±.02 | **0.50±.01** | **0.50±.01** | 0.32±.01 | 0.30±.01 | 0.30±.01 |
| CFR MMD$_{com}$ | **0.55±.01** | **0.50±.01** | **0.50±.01** | 0.32±.01 | 0.30±.01 | 0.30±.01 |
| CEVAE$_{com}$ | **0.39±.01** | 0.37±.01 | 0.37±.01 | **0.08±.02** | **0.05±.01** | **0.05±.01** |
| FedCI | **0.41±.01** | **0.40±.01** | **0.39±.01** | **0.05±.01** | **0.04±.01** | **0.03±.01** |

Table 4: Out-of-sample errors on count outcomes data.

| Method | The error of ITE ($\sqrt{\epsilon_{PEHE}}$) | | | The error of ATE ($\epsilon_{ATE}$) | | |
|---|---|---|---|---|---|---|
| | 1 source | 3 sources | 5 sources | 1 source | 3 sources | 5 sources |
| BART$_{com}$ | 6.30±.06 | 6.29±.04 | 6.26±.03 | 0.75±.14 | 0.59±.18 | 0.47±.13 |
| X-Learner$_{com}$ | 6.10±.10 | 5.16±.06 | 4.72±.03 | 1.34±.29 | 0.63±.12 | 0.42±.08 |
| R-Learner$_{com}$ | 6.27±.06 | 6.09±.05 | 5.89±.04 | 0.82±.13 | 0.66±.15 | 0.56±.10 |
| OthoRF$_{com}$ | 6.02±.29 | 4.15±.06 | **3.74±.05** | 0.75±.18 | 0.54±.17 | **0.41±.10** |
| TARNet$_{com}$ | 4.54±.14 | **3.98±.05** | 3.80±.02 | 0.77±.10 | 0.66±.02 | 0.62±.03 |
| CFR Wass$_{com}$ | **4.08±.04** | 4.03±.03 | 3.78±.02 | 0.72±.04 | **0.51±.03** | 0.50±.03 |
| CFR MMD$_{com}$ | 4.15±.06 | 4.05±.04 | 3.77±.02 | **0.69±.07** | 0.54±.03 | 0.50±.03 |
| CEVAE$_{com}$ | **3.40±.09** | **3.31±.07** | 3.08±.05 | 0.56±.16 | 0.40±.12 | 0.35±.08 |
| FedCI | **4.02±.10** | **3.05±.08** | **2.66±.04** | 0.54±.09 | 0.48±.08 | **0.25±.05** |

Table 5: Out-of-sample errors on IHDP dataset. Please see the full table in Appendix F.2.

| Method | The error of ITE ($\sqrt{\epsilon_{PEHE}}$) | | | The error of ATE ($\epsilon_{ATE}$) | | |
|---|---|---|---|---|---|---|
| | 1 source | 2 sources | 3 sources | 1 source | 2 sources | 3 sources |
| BART$_{com}$ | 5.98±2.7 | 4.32±2.1 | 4.04±2.0 | 1.80±1.1 | 2.09±1.1 | 1.21±0.6 |
| X-Learner$_{com}$ | 4.22±1.6 | 4.15±1.5 | 4.06±1.8 | 1.64±0.7 | 1.93±0.8 | 0.84±0.4 |
| R-Learner$_{com}$ | 6.97±2.1 | 4.43±1.4 | 4.47±1.7 | 3.15±0.5 | 1.34±0.5 | 1.10±0.3 |
| OthoRF$_{com}$ | 4.49±1.9 | 3.81±1.3 | 3.75±1.5 | 1.86±0.8 | 1.61±0.6 | 1.56±0.8 |
| TARNet$_{com}$ | 4.50±1.4 | 3.15±0.8 | 3.79±1.1 | **1.52±0.5** | 1.18±0.4 | 0.91±0.3 |
| CFR Wass$_{com}$ | **4.37±1.2** | 2.93±0.6 | 2.85±0.9 | **1.18±0.7** | **0.72±0.2** | 0.67±0.1 |
| CFR MMD$_{com}$ | 4.43±1.3 | **2.85±0.6** | 2.83±1.1 | 2.32±0.8 | **0.63±0.2** | 0.54±0.2 |
| CEVAE$_{com}$ | **3.16±0.6** | 2.34±0.6 | 2.31±0.7 | 2.02±0.4 | **0.53±0.1** | 0.48±0.2 |
| FedCI | **2.88±0.8** | **2.36±0.5** | **2.35±0.6** | **1.43±0.7** | 1.03±0.4 | **0.51±0.2** |

development. Herein, specialist visit is the treatment and children's cognitive development is the outcome. We use the NPCI package [Dorie, 2016] to simulate two potential outcomes for the treatment (with or without specialist visit) of each child. Hence, the *true* individual treatment effect can be computed for evaluation purposes. There are 10 replicates of the dataset, and each of them is divided into three sources of size 249. For each source, we then split it into three equal sets for the purpose of training, testing, and validating the models. The mean and standard error of the aforementioned evaluation metrics are reported over the above 10 replicates of the data.

**Results and discussion.** Similar to the experiment for synthetic datasets, here we also train the baselines in three cases as explained earlier. We also expect that the errors of FedCI to be close to the baselines trained with combined data (com). The results reported in Table 5 show that FedCI achieves competitive results compared to the baselines (we skipped the first and second cases (loc and agg), please see Appendix F.2 for the full table). Indeed, FedCI is in the top-3 performances among all the methods. This result again verifies that FedCI can be used to estimate causal effects effectively under privacy-perserving, federated data settings. The estimated distribution of ATE is presented in Appendix F.2 due to limited space.

## 5 CONCLUSION

We have introduced FedCI, a Bayesian causal inference paradigm via a reformulation of multi-output GPs to learn causal effects, while keeping data at their local sites. An inference method involving the decomposition of ELBO is presented, allowing the model to be trained in a federated setting.

This work is an important step towards a privacy-preserving causal learning model. One interesting future research direction is to combine FedCI with differential privacy to give a stronger privacy guarantee. This new direction would require adding an appropriate noise component, such as Laplace noise, to the parameters while training the model. Our future research would also involve further exploration on combining FedCI with differential privacy for Gaussian processes [Smith et al., 2018] and multiparty differential privacy algorithms [Pathak et al., 2010, Rajkumar and Agarwal, 2012, Hamm et al., 2016].

The inherent use of GPs in our approach would incur computational time of inverse covariance matrix in each source of cubic time complexity. Another possible future research direction is to reformulate this in terms of sparse Gaussian process models [Hensman et al., 2013, Hoang et al., 2017, 2020].

## Acknowledgements

This research/project is supported by the National Research Foundation Singapore and DSO National Laboratories under the AI Singapore Programme (AISG Award No: AISG2-RP-2020-016).

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
