# OpenReview forum: "Bayesian Federated Estimation of Causal Effects from Observational Data"
_auai.org/UAI/2022/Conference — UAI 2022 Poster_

### Official Review · Reviewer_HFQJ · 2022-03-29

**Q2(1) Originality/Novelty:** 2
**Q2(2) Significance/Impact:** 2
**Q2(3) Correctness/Technical Quality:** 3
**Q2(6) Clarity Of Writing:** 3
**Q6 Overall Score:** 5
**Q8 Confidence In Your Score:** 3

**Q1 Summary And Contributions:**

The authors present a Bayesian approach to estimating causal effects in a federated setting.

The main contribution is that the authors' approach takes into account the constraint that data from multiple sources cannot be combined or shared, but also retains competitive performance against methods which do not satisfy this constraint.

**Q2 Assessment Of The Paper:**

More detailed information regarding each of these aspects is given below:

**Q2(4) Quality Of Experiments (Optional):**

3: Good: The experimental evaluation is adequate, and the results convincingly support the main claims.

**Q2(5) Reproducibility:**

3: Good: Key resources (e.g., proofs, code, data) are available and key details (e.g., proofs, experimental setup) are sufficiently well-described for competent researchers to confidently reproduce the main results.

**Q3 Main Strengths:**

The authors explain the problem setting and their approach in a clear and comprehensive manner. The paper is easy to follow, because the authors start from simpler concepts and gradually build toward the more complicated subject matter in a natural way.

The assumptions made in their problem formulation are made explicit and they are well justified.

Experiments are made with both real and synthetic data, and compared to a variety of other methods, even in a disadvantageous scenario where the core privacy constraint was violated.

**Q4 Main Weakness:**

The approach of the authors seems quite limited in terms of practical applicability. While they propose flexibility via the inter- and intradependency components in their model,  the responses themselves are still gaussian and scalar. I wonder how the proposed approach could be applied for count or ordinal data. Also, equations 2 and 3 immediately rule out mediation and more general treatment-to-outcome causal paths.

Since variational approximation is used, the bias of the estimates should be studied and reported.

Robustness of the proposed method in terms of model misspecification was not explored.

Scalability of the proposed method is not discussed, neither in terms of computation time. sample size or number of sources or studied via simulations.

**Q5 Detailed Comments To The Authors:**

The authors dismiss transportability outright, stating that it "does not take into account above data privacy constraint". While it is true that there is no explicit consideration in transportability algorithms with regards to this constraint, I believe the authors could have explored this a bit more. In general, if a transportability algorithms identifies a causal effect from multiple sources, it combines the estimands, which would violate the constraint, but, it is not necessary to do so. One could simply choose the estimand which had the smallest variance etc. thus preserving the privacy constraint (i.e., only summary statistics would be combined).

I agree with the authors that Assumption 3 is reasonable, but they could also discuss the case where it is violated at greater length, e.g., by modelling the missing covariates along with the response variables at each site to inpute the missing data.

I would replace the capital phi symbol with something else, since many readers may associate it with the CDF of a standard normal distribution

**Q7 Justification For Your Score:**

While the presentation is mostly clear and free of any apparent technical flaws, I'm not convinced of the overall contribution of the proposed approach. In my opinion, the practical applicability of the proposed approach is somewhat limited, That being said, I'm not an expert in federated learning.

**Q9 Complying With Reviewing Instructions:**

1: Yes.

---

### Official Review · Reviewer_Pqy3 · 2022-04-03

**Q2(1) Originality/Novelty:** 3
**Q2(2) Significance/Impact:** 3
**Q2(3) Correctness/Technical Quality:** 3
**Q2(6) Clarity Of Writing:** 3
**Q6 Overall Score:** 7
**Q8 Confidence In Your Score:** 3

**Q1 Summary And Contributions:**

The paper proposes a method based on Gaussian Processes for the estimation of ATE from federated data, that is, when data cannot be stored in a single place due to privacy constraints. The experimental results show that the method is competitive with previous ATE estimation methods, which do not respect the privacy constraints.

**Q2 Assessment Of The Paper:**

More detailed information regarding each of these aspects is given below:

**Q2(4) Quality Of Experiments (Optional):**

3: Good: The experimental evaluation is adequate, and the results convincingly support the main claims.

**Q2(5) Reproducibility:**

4: Excellent: Key resources (e.g., proofs, code, data) are available and key details (e.g., proof sketches, experimental setup) are comprehensively described for competent researchers to confidently and easily reproduce the main results.

**Q3 Main Strengths:**

The paper is strong in most areas. It is novel (although maybe not revolutionary) and highly significant based on the experimental results. It is clearly written and uses a precise but readable language. The code is provided for reproducibility.

**Q4 Main Weakness:**

Two minor weaknesses of the work are:

(i) The concept of ATE with different sources of data is not presented in sufficient detail in the paper. The ATE is estimated according to the number of elements for each site source of data, providing more weight to those sources for which a larger amount of data has been gathered. Many times, the percentages of sampling in different sources are different. Thus, the data does not reflect the proper distribution of population between the sources. The paper lacks an assumption of "homogeneity" of the sampling rates at the different sources of data.

(ii) I think that a more thorough comparison on other real world datasets would improve the experimental section.


**Q5 Detailed Comments To The Authors:**

Can you discuss about "homogeneity". Is this assuption made somewhere in the paper?

**Q7 Justification For Your Score:**

The paper presents a state of the art ATE method which is able to maintain the privacy of the data coming from different sources. Thus, the paper is quite strong in all evaluable aspects. Both weaknesses identified are minor.

**Q9 Complying With Reviewing Instructions:**

1: Yes.

---

### Official Review · Reviewer_HFYK · 2022-04-11

**Q2(1) Originality/Novelty:** 3
**Q2(2) Significance/Impact:** 3
**Q2(3) Correctness/Technical Quality:** 3
**Q2(6) Clarity Of Writing:** 2
**Q6 Overall Score:** 7
**Q8 Confidence In Your Score:** 4

**Q1 Summary And Contributions:**

This paper proposes a Federated Causal Inference (FedCI) framework that fuses federated learning and causal inference to incorporate multiple data sources while maintaining the sources at their local sites. The proposed framework uses the Bayesian approach to estimate the causal effect. The idea is new.

**Q10 Ethical Concerns (Optional):**

There is no ethical concern.

**Q2 Assessment Of The Paper:**

More detailed information regarding each of these aspects is given below:

**Q2(4) Quality Of Experiments (Optional):**

3: Good: The experimental evaluation is adequate, and the results convincingly support the main claims.

**Q2(5) Reproducibility:**

2: Fair: Key resources (e.g., proofs, code, data) are unavailable but key details (e.g., proof sketches, experimental setup) are sufficiently well-described for an expert to confidently reproduce the main results.

**Q3 Main Strengths:**

1. The paper proposes a new idea about how to estimate the causal effect from federated observational data sources using the Bayesian approach.

2. The paper provides  theoretical support for the proposed method.

3. Experiments show that the proposed method has good performance through comparing with several other methods.

**Q4 Main Weakness:**

1. There are some minor errors in the paper.

2. The meaning of some notations and some procedure in Algorithm 1 need to be explained.

**Q5 Detailed Comments To The Authors:**

1. In the introduction section, it says that "For example, a narrow confidence interval for individual treatment effect of smoking on lung cancer means that the patient is at a higher risk of getting cancer."  This statement may not be true in all situations.  If zero falls into the confidence interval for the individual treatment effect and the confidence interval is narrow, then smoking does not have significantly effect on the lung cancer and the statement is wrong.

2. The meaning of some notations needs to be explained. For example, the meaning of D_{KL} in the formula of L^s.

3. In algorithm 1, it needs to compute the derivative of L^s, can you explain more about how to compute it?

4. In experiments, based on Figure 1, the error of ATE for the FedCI method increases as the number of sources increases from 1 to 2. Based on Figure 2, the error of ATE for the FedCI method is larger than that for FedCI without inter-dependency, can you explain the reasons for these results?

5. There are some typo errors. For examples, in formulas (12) and (13), the sign s is ignored in some notations.


**Q7 Justification For Your Score:**

I make this score based on the main strengths,  weaknesses and my understanding of this paper. I think the strengths of this paper far outweigh its weaknesses.

**Q9 Complying With Reviewing Instructions:**

1: Yes.

---

### Official Review · Reviewer_iFBg · 2022-04-15

**Q2(1) Originality/Novelty:** 3
**Q2(2) Significance/Impact:** 3
**Q2(3) Correctness/Technical Quality:** 3
**Q2(6) Clarity Of Writing:** 2
**Q6 Overall Score:** 7
**Q8 Confidence In Your Score:** 3

**Q1 Summary And Contributions:**

This paper considers Bayesian treatment on the estimation of causal effect under federated setting where the data sets across multiple sites cannot be shared directly. The building block of the proposed estimator is combining i) Gaussian process for outcome model ii) capturing intra and inter dependency across data sources iii) variational approach to learning federated objective. The resulting estimator works well on synthetic data sets widely used in the community.

**Q2 Assessment Of The Paper:**

More detailed information regarding each of these aspects is given below:

**Q2(4) Quality Of Experiments (Optional):**

3: Good: The experimental evaluation is adequate, and the results convincingly support the main claims.

**Q2(5) Reproducibility:**

4: Excellent: Key resources (e.g., proofs, code, data) are available and key details (e.g., proof sketches, experimental setup) are comprehensively described for competent researchers to confidently and easily reproduce the main results.

**Q3 Main Strengths:**

- The paper neatly combines essential ideas to work on the federated setting.
- Experiments are done with various settings to examine the quality of the proposed algorithm.
- The paper is well-motivated.

**Q4 Main Weakness:**


No clear major weakness but some minor weaknesses, which I will explain in Q5.

**Q5 Detailed Comments To The Authors:**

Comparing to transportability, the paper assumes that the different sources represent no difference at all (Eq 9). Then, when proposing (Eq 9) for specific sources, there exist correlation between two outcomes and between multiple sources.

The paper does not justify the use of such dependency at all and just proceed to propose to model such dependency. Surely, if no dependency does not exist, then learned parameters will be simply exhibit independence. So could you explain a little bit more on the reason for "To capture dependency among the sources" and "matrices capturing the correlation between the possible outcomes"

Figure 2 is explained very little. Why is it the case that No-interdependency work very differently between PEHE and ATE?  I can see the interdependency helps. But again, as mentioned above, please provide why it is needed in the first place.

Regarding comparisons, BART is really a baseline and CEVAE is not the state of the art. Simple NN-based model (TARNet, CFR, Dragonnet) would be good to compare (while they are not for the federated learning!)

Q. baselines (3) in aggregation, do you combine based on minimum variance weighting or just simple averaging?

Some minor weaknesses

- Related work on causal inference is relatively weak in that it focuses more on some of the recent techniques.
- The paper is written a bit densely focusing on the math (even with proofs moved to appendix). It would be greatly appreciated if a simple diagram representing how learning works is presented early around Section 3.
- Section and subsection names are not so descriptive (e.g., out approach, proposed model, proposed algorithm ...)

====
Several concerns are cleared up.

**Q7 Justification For Your Score:**

The novelty can be viewed differently among the reviewers since the idea of Bayesian, GP, federated learning, Potential Outcome framework, Variational Inference are all well-defined and studied. Yet the authors put those things correctly and showed the merit of such approach.


**Q9 Complying With Reviewing Instructions:**

1: Yes.

---

### Decision · Program_Chairs · 2022-05-15

**Decision:**

Accept (Poster)

**Comment:**

Meta Review: The paper considers Bayesian estimation of causal effects in a federated setting where data is at multiple sites and cannot be directly shared.

Pros:
-This setting seems to be novel.

-Experimental evaluation is thorough and favourable.

-Combines Bayesianism, Gaussian processes and variational inference.

Cons:

-Minor unclarities in writing.

Reviewers all suggest acceptance, the recommendation is an accept.